



# Timescales of emergence of chronic nuisance flooding in the major economic centre of Guadeloupe.

Gonéri Le Cozannet[1], Déborah Idier[1], Marcello de Michele[1], Yoann Legendre[2], Manuel Moisan[2], Rodrigo Pedreros[1], Rémi Thiéblemont[1], Giorgio Spada[3], Daniel Raucoules[1], Ywenn de la Torre[2]

[1]BRGM, DRP/R3C, Orléans, 45000, France
[2]BRGM, DAT/GUA, Petit-Bourg, 97170, France
[3]Dipartimento di Scienze Pure e Applicate (DiSPeA), Università di Urbino "Carlo Bo", Italy

*Correspondence to*: Gonéri Le Cozannet (g.lecozannet@brgm.fr)

**Abstract.** Chronic flooding, occurring at high tides under calm weather conditions, is occasionally taking place today in the low-lying areas of the Petit-Cul-de-sac marin (Guadeloupe, West Indies, French Antilles). This area includes critical industrial, harbor and major economic infrastructures for the island. As sea level rises, concerns are growing regarding the possibility for

repeated chronic flooding events, which would alter the operations at these critical coastal infrastructures without appropriate adaptation. Here, we use information on past and future sea levels, vertical ground motion and tides to assess times of emergence of chronic flooding in the Petit-Cul-de-sac marin. For RCP8.5 (i.e., continued growth of greenhouse gas emissions), the number of flood days is projected to increase rapidly after the emergence of the process, so that coastal sites will be flooded every two days within 2 decades after the onset of chronic flooding. For coastal locations with the smallest altitude, we show

that the reconstructed number of floods are consistent with observations known from a previous survey. One key uncertainty of our result is the actual rate of subsidence of the island. However, our satellite interferometric synthetic-aperture radar results show that the local variability of this subsidence is smaller than the uncertainties of the technique, which we estimate between 1 (standard deviation of measurements) and 5mm/yr (upper theoretical bound). Our results imply that adaptation pathways considering a rapid increase of recurrent chronic flooding are required in the critical port, industrial and commercial center of

Guadeloupe, as well as presumably in many low-elevation coastal zones of other tropical islands.

Keywords: Chronic flooding, sea-level rise, subsidence, tides, nuisance flooding.


## 1 Introduction

Chronic flooding is defined as flooding occurring at high tides under calm weather conditions (Ezer and Atkinson, 2014;Sweet and Park, 2014;Moftakhari et al., 2015;Moftakhari et al., 2017;Dahl et al., 2017). Since 30 years, reports from the Intergovernmental Panel on Climate Change (IPCC) have shown that mean sea level is rising due to anthropogenic climate warming, and that this will continue in the future (Church et al., 2013a;Garner et al., 2018;Oppenheimer et al., 2019). Therefore, risks of chronic flooding are expected to rise in an increasing number of locations in the future, including

embankments in harbors, cities and settlements in sheltered areas and other regions where extreme events are rare enough to justify low protection levels. Furthermore, risks of chronic flooding are already taking place now: for example, an acceleration in chronic flooding is already affecting coastal communities and transportation systems in the US eastern coast (Sweet and Park, 2014;Jacobs et al., 2018). There, the observed trends and acceleration of the number of chronic flooding events is consistent with increases in mean sea level caused by climate change (Sweet and Park, 2014). Among coastal impacts of sea-

level rise, risks of chronic flooding require the most immediate adaptation response, in particular in terms of coastal defenses upgrades and water networks management (Le Cozannet et al., 2017). Today, the topic is becoming even more impelling due to the observed acceleration of sea-level rise caused by the onset of Greenland and Antarctica ice-sheets melting (Dangendorf et al., 2017;Dieng et al., 2017;Chen et al., 2017).

Despite the relative urgency of the topic, there is still little awareness regarding chronic flooding and their potential impacts

in a number of regions. Here we focus on a tropical island, Guadeloupe, located in the Lesser Antilles in the Eastern Caribbean. Guadeloupe is among the small islands where current flooding risks are primarily associated to the seasonal occurrence of tropical cyclones (Krien et al., 2015;Rueda et al., 2017). Guadeloupe presents a complex and steep topography due to its volcanic origin, but existence of low-lying areas, in particular in Petit Cul-de-sac marin, with cities of Pointe-à-Pitre et Baie-Mahault, have allowed the recent development of critical infrastructure such as power plants, port and airport facilities and

commercial areas, which have been built on a former low-lying mangrove (Bourdon and Chiozzotto, 2012). This situation of high exposure to coastal hazards in low-lying area is typically representative of the current situation in many inhabited high-islands in tropical regions (Nurse et al., 2014). Bourdon and Chiozzotto (2012) reported chronic flooding events as well as damages to concrete infrastructures due to salinization of groundwater. However, neither the link with climate-induced sea-level changes nor the timescales of emergence of recurrent chronic flooding have been characterized yet, with more attention

being given on extreme events such as cyclones so far (Krien et al., 2015;Jevrejeva et al., 2020). For critical infrastructures, the current frequency of chronic flooding is not affecting activities yet. However, this could be different in the future as sea level rises.

In this contribution, we aim at characterizing the timescales of emergence of chronic flooding in the area of the Petit-Cul-de-sac marin, which is a hotspot for systemic vulnerability in Guadeloupe (Figure 1; section 2). To do so, we quantify the different

phenomena causing relative present and future mean sea-level changes, on the one hand, and the high-water levels resulting from tides, surges and mean sea-level fluctuations induced by regional circulations, excluding cyclone events on the other





hand (section 3). In section 4, we present the projected timescales of emergence of recurrent chronic flooding for different scenario and coastal settings. These results come with residual uncertainties, such as the assumption that sea-level rise does not modify tidal maxima. These residual uncertainties are discussed in section 5, together with the significance of our results for the attribution of chronic flooding events and for coastal management. Overall, our results illustrate that the question of chronic flooding could deserve some attention in tropical islands concerned with adapting to the future effects of sea-level rise.

## 2 Coastal Settings

This study focuses on the coasts of the Petit-Cul-de-sac marin, a large low-lying area located at the junction of Basse-Terre and Grande-Terre, the two main islands of Guadeloupe, Basse-Terre and Grande-Terre, which are separated by a tight channel, the Rivière salée (Figure 1). This sheltered coastal area is characterized by large mangrove areas, a reef barrier and active sedimentation processes. The area is recognized as being highly exposed to marine flooding due to cyclones and tsunamis (Pedreros et al., 2016). However, chronic flooding has received less attention so far, despite the fact that previous studies have extensively inventoried the most vulnerable sites with respect to this process.

The low-lying areas of the Petit-Cul-de-sac marin have been intensively urbanized over the last century. The major assets at risk include:

- The industrial zone of Jarry (Baie-Mahault), located on the east side of the island of Basse-Terre: this area includes harbor, industrial, energy production, commercial and storage facilities, which collectively constitute the major economic center of Guadeloupe.
- The landing area of the international airport in Guadeloupe (Les Abymes).
- The historical center of the city of Pointe-à-Pitre: this area includes residential buildings and the main ferry terminal of Guadeloupe.
- Other residential buildings in the northern districts of the Pointe-à-Pitre area (Lauricisque and Raizet, Figure 1)

Mangrove areas have been largely reduced due to the urbanization since the 1950s (Figure 2). In addition, 85 hectares of land have been reclaimed from the sea to facilitate harbor activities (Roques et al., 2010). Land reclamation projects are still being investigated to further support the development of the port of Guadeloupe.

Surveys have shown that the highest water levels are extending in the area of the Petit-Cul-de-sac marin (Bourdon and Chiozzotto, 2012): more frequent chronic flooding is reported by interviewees in some of the reclaimed areas, and some local areas seems more often saturated with water. This perception of rising coastal water levels is confirmed by observations, such as soil compaction affecting roads built on former mangroves, which could be due to higher groundwater levels reducing the load-bearing capacity of the backfill according to Bourdon and Chiozzoto (2012). Other signs of rising coastal water levels include damages to concrete infrastructures due to the salinization of ground waters, the chronic flooding of the lowest zones, the destabilization of some structure foundations and damages to the sewage system. However, some of these impacts can also be caused by heavy rains (Pedreros et al., 2016).



Fieldwork undertaken by Bourdon and Chiozzoto (2012) has allowed mapping the most vulnerable hotspots for chronic

flooding in the Petit-Cul-de-sac marin area (Figure 1). These include:

- land area located close to the Rivière Salée, the channel located between the two islands

- mangrove areas located between Gabarre and Le Raizet (Pointe-à-Pitre)

- the international airport landing area (Les Abymes)

- a large part of the districts of Lauricisique, Bergevin (Pointe-à-Pitre) and some areas at Grand-Camp (Les Abymes)

- areas bounding Morne à Savon close to Jarry (Baie-Mahault)

- a large part of the port embankments in Jarry (Baie-Mahault)

To conclude, the observations and surveys above suggest that relative sea-level rise is causing chronic flooding and other damages in Petit Cul-de-sac marin and in particular in Jarry (Baie-Mahault) and Pointe-à-Pitre area. This is of major significance for the region, due to the critical economic role of this area for the entire Guadeloupean archipelago.

**3 Data and Methods**

**3.1 From global to regional sea-level rise**

Contemporary global sea-level rise is due to ocean thermal expansion, the melting of mountain glaciers and ice-sheets, and contributions from land water (Stammer et al., 2013). However, regional sea level changes in the Caribbean differ from the global mean due to (1) sterodynamic sea level changes due to changes in the ocean density and circulation and inverse

barometer effects, and (2) the changes in Earth gravitation, rotation and elastic response associated to current ice or water mass redistributions, as well as the viscoelastic response of the solid Earth to the last deglaciation (Glacial Isostatic Adjustment, GIA) (Gregory et al., 2019). In this subsection, we detail the data and methods used for evaluating past and future sea-level changes in Guadeloupe.

*Past sea-level changes*

The longest tidal record in Guadeloupe is short (19 years), which prevents from a thorough characterization of past sea-level change in this region. To complete this data, we use sea-level reconstructions from 1950 to 2010 of Meyssignac et al. (2012), which have been extensively analyzed in the Caribbean (Palanisamy et al., 2012). Sea-level reconstructions use long (but pointwise) sea-level change records from tide gauges with information from satellite altimetry or ocean dynamic models  to provide spatial maps of sea-level trends and variability (Meyssignac et al., 2012). While sea-level reconstructions have

limitations due to the lack of long term in-situ sea-level records for validation in some regions, those from Meyssignac et al. (2012) perform best in representing past sea-level changes and variability, and therefore represent the state of the art in this area (Carson et al., 2017). They use the spatial patterns observed in the altimetry era on the one hand, and those obtained from two ocean models Soda and Drakkar on the other hand, thus resulting in three reconstructions, referred to as "Altimetry", "Drakkar" and "Soda" hereafter.





*Future sea-level changes: likely range*

We consider sea-level scenario based on RCP8.5, which assumes either an increase of greenhouse gas emissions over the height remaining decades of the 21$^{st}$ century (Riahi et al., 2017), or significant carbon emissions from the permafrost (Meredith et al., 2019). We do not consider other climate change scenarios such as RCP4.5 or RCP2.6, because they imply that all critical infrastructures related to energy and transport will undergo a major conversion over the next decade (Rockström et al., 2017).

Yet, we do not know how future energy and transportation infrastructures will look like after such a transformation, making any assessment of their vulnerability highly speculative.

Sea-level projections from the 5th report of the International Panel of Climate Change (AR5 IPCC) (Church et al., 2013a) are available from the Integrated Climate Data Center of the University of Hamburg ([http://icdc.cen.uni-hamburg.de/](http://icdc.cen.uni-hamburg.de/)) (Carson et al., 2016). However, the Antarctic contribution has been updated in the recent IPCC special report on Ocean and Cryosphere

in a changing climate (SROCC). These projections sum up the regional effects of each component of future sea-level rise (Slangen et al., 2012;Slangen et al., 2014;Gregory et al., 2019), that is, sterodynamic effects, the melting of mountain glaciers and ice sheets, the contribution of land water and GIA. These projections do not include potential additional local vertical ground motion (subsidence or uplift) due to regional to local natural or anthropogenic effects, which we evaluate in subsections 3.2 and 4.2 below. For the sterodynamic effects, we use the same climate models as in the IPCC AR5 and SROCC reports,

excluding the MIROC-ESM and MIROC-ESM-CHEM climate models, which are outliers in this region (Thiéblemont et al., 2019). For all other components, we use the median and likely projections of the latest SROCC report (Oppenheimer et al., 2019) and compute the regional effects in Guadeloupe using the regionalization method of Slangen et al. (2012) using fingerprints for mass contributions.

*Future sea-level changes: high-end scenario*

According to the SROCC, there remains a probability of 33% for sea-level rise to lies outside the likely range. While the low bound of the likely range can be considered a minimum sea-level commitment (Le Cozannet et al., 2019), there remains the possibility of unlikely, but possible scenarios above the likely range(Stammer et al., 2019). The largest risks of exceeding the IPCC likely range are related to the melting of ice-sheets in Antarctica and Greenland, which involve processes that are not yet fully understood (Furst et al., 2015;DeConto and Pollard, 2016;Hanna et al., 2018;Pattyn, 2018;Edwards et al.,

2019;Oppenheimer et al., 2019). These high-end scenarios are increasingly considered as relevant information for the most risk averse users (Nicholls et al., 2014;Hinkel et al., 2015;Hinkel et al., 2019), which would be presumably the case for the port, industrial and commercial facilities of our case study.

So far, IPCC authors have refused to provide projections beyond the likely range, considering that existing results were not robust enough yet (Church et al., 2013b). However, since the AR5, other studies have proposed high-end scenarios (Jackson

and Jevrejeva, 2016;Le Bars et al., 2017;Kopp et al., 2017;Thieblemont et al., 2019). Here, we use the global assumptions of Thiéblemont et al. (2019) (their "high-end B"), which follows a "worst-model" approach, that is, not necessarily the upper limit to sea-level rise, but upper values available from the literature (Table 1). As high-end projections of each sea-level component provided in Thiéblemont et al. (2019) are estimated for the year 2100 only, we produce annual time series over the





21$^{st}$ century by fitting a spline function to interpolate between the recent past (2007-2020 period) and 2100. Then, we
regionalize these values following again the well-established approach of Slangen et al. (2012).

## 3.2 Regional and local vertical ground motion

Relative sea-level changes at the coast may be affected by regional and local vertical ground motion (Raucoules et al., 2013;Woppelmann and Marcos, 2016;Martinez-Asensio et al., 2019). In the context of this study, regional vertical ground motion may be due to regional tectonic processes, whereas a local subsidence could be caused by changes in the water content
of artificialized ground, causing sediment compaction.

*Regional vertical ground motion*

We use different lines of evidence to characterize regional vertical ground motion taking place at the scale of the island. The first sources of information available are measurements from four permanent GNSS stations located close to Jarry (Baie-Mahault) (Table 2; Figure 4). For each of these stations, we use two solutions from the Nevada Geodetic Laboratory and from
SONEL ([www.sonel.org](www.sonel.org)) to evaluate potential subsidence or uplift trends at these stations (Santamaria-Gomez et al., 2017;Blewitt et al., 2018). Other GNSS stations are available, but they are not considered here either because they are too short (e.g., North Grande Terre), or because they are too close to the crater of the active volcano of La Soufrière, on the south part of the western island (Basse-Terre).

The second source of information are sea-level time series at the tide gauge of Pointe-à-Pitre. This information can be used to
further upraise vertical ground motion, either combining it with regional sea-level trends from satellite altimetry (Cazenave et al., 1999), or by analyzing the different modes of spatial and temporal variability in sea-level time series (Kopp, 2013). The first technique comes with large uncertainties (Ablain et al., 2015;Le Cozannet et al., 2015;Woppelmann and Marcos, 2016). However, it can still be useful to identify or reject very fast coastal vertical ground motion, in the order of centimeters per years, as shown for example in the case of Manila in the Philippines (Santamaria-Gomez et al., 2012;Raucoules et al., 2013).
Observations of vertical ground motion with GNSS stations include the effects of the GIA, which are also included in regional sea-level rise projections discussed in section 3.1.

Overall, the tide gauge measurements are short (19 years; see subsection 3.3) and the GNSS data include discontinuities, presumably due to instrumentation changes or earthquakes. Hence, we do not expect to obtain a single reliable assessment, but instead, we use these lines of evidence to design contrasting scenarios for future vertical ground motion.
*Local vertical ground motion*

To characterize local vertical ground motion, we use Synthetic Aperture Radar (SAR) interferometry. This technique is increasingly being used in various fields of the Earth Sciences since the 1990s to measure and detect the deformation of the ground surface (Gabriel et al., 1989;Massonnet et al., 1993). The SAR is an "active" system, which sends out its own source of illumination, emitting waves in the microwaves field of the electromagnetic spectrum. A first antenna emits a signal (wave
beam), which is then recorded by another or the same antenna after its backscattering on the Earth's surface. As the atmosphere



and the clouds are almost transparent in this range of wavelengths, the SAR instruments can acquire images at day and night, regardless of the meteorological conditions. The possibility of a continuous illumination of the radar and the orbital characteristics of the satellites makes it possible to distinguish two geometric configurations for the acquisitions: the so-called "descending" orbits, during which the satellite moves approximately from the northeast to the southwest, and "ascending" orbits, during which the satellite moves from south-east to north-west.

In this study, we used 34 images of the Advanced Synthetic Aperture Radar (ASAR) sensor of the European Envisat satellite acquired between 19/01/2003 and 10/01/2010. The images belong to track / frame 75/315 in an ascending orbit (de Michele, 2010). All possible interferometric couples are calculated from 34 SAR data. We keep only the differential interferograms characterized by a short perpendicular baseline, which determines the sensitivity of the signal to the topography and impacts the interferograms' quality due to geometric decorrelation. Thus, we will intrinsically use the interferograms that are not (or little) affected by the topography. We set the threshold of baselines perpendicular to 150 meters. Thus, 65 differential interferograms with short baseline are calculated.

To exploit the entire InSAR database, we use a processing method known as Small Baseline Subset or SBAS (Berardino et al., 2002). This method, developed by Usai et al. (2003) is implemented in the GAMMA tool chain (from Gamma Remote Sensing AG) under the name of multi-Baseline (Wegmüller et al., 2009). As a result, we produce a map of linear velocities, measured along the line of sight of the satellite, that is, making an angle of 23° on the vertical.

### 3.3 Total water levels causing chronic flooding

We compute the daily high water levels (one value/day) using the hourly water level measurements of a tide gauge located at Pointe-à-Pitre (tide gauge location: -61.5315°E; 16.2244°N; doi:10.17183/REFMAR#125). These measurements are provided by the French Hydrographic Service (SHOM) and available on the Refmar database (data.shom.fr). In June 2017, these data were covering about 19 years of useful data, starting the 4th of January 1983). The observed tidal signal includes effects of the following phenomena: tides (tidal range up to about 40 cm), mean sea-level seasonal variations related to oceanic circulations (ranging between about 10 and 30 cm depending on the years) and storm surges, either caused by tropical storms or cyclones. Here we aim at characterizing highest water levels per day, relative to mean sea level and representative of moderate conditions, that is, excluding effects of cyclone events. To do so, we proceed as follows: after a quality check and cleaning of the data, we identify 20 cyclones over the study period, representing 106 days using IbTracks data (Knapp et al., 2010), together with Météo-France information. These data are removed from our dataset. Then, we keep only the years with a completeness of 90%, in order to properly account for mean sea-level seasonal variations. We also keep only days with no gaps in order to properly account for the tide fluctuations in the distribution. The final dataset covers 16 years (1983, 1991 to 1997, 2006, 2007, 2010 to 2014, 2016 ; each covered at more than 90%). Then, we detrend the data from observed sea-level rise at the tide gauge, over the period of observations (0.7 mm/y). Finally, we extract the daily maxima from these dataset and reference them





vertically with respect to the terrestrial vertical datum IGN88, for a mean sea level corresponding to the ones of the 31rd of December 2016.

**3.4 Exposure of coastal sites**

We evaluate the altitude of 34 priority coastal sites identified as vulnerable to chronic flooding by Boudon and Chiozzoto (2012). No precise measurement of the altitude is given in this report, but we benefit from a recent LiDAR measurements in Guadeloupe  (LITTO3D© - IGN & SHOM) to estimate their altitude. This comes with the limitations of the exact location of each site, with a few meter of geolocation errors potentially accounting for different pixels in the LiDAR maps, and therefore errors in the evaluation of the altitude. The LITTO3D© dataset itself has typically errors in the order of 20cm vertically.

Bourdon and Chiozzoto (2012) classified the coastal sites in three categories: low (8 sites), medium (13 sites) and high (13 sites) vulnerability to chronic flooding, based on multi-parameter analysis on the field and surveys (Figure 1).

**3.5 Synthesis: evaluation of past and future chronic flooding**

We compute the number of nuisance flooding days per year as the sum of projected mean sea-level changes with the daily maxima of water levels resulting from tide, surges and mean seasonal variations related to the regional circulations. As we are

interested in chronic flooding, we exclude the water levels resulting from cyclone events in the present study. Our approach neglects tide-surge interactions, and we also note that although the coastline is largely artificialized and stabilized, the area is mostly free from flood defenses such as small walls or dike that could prevent chronic flooding to take place.

**4 Results**

**4.1 Regional sea-level rise**

Geocentric sea-level reconstructions projections for Guadeloupe are presented in Figure 3.A. The reconstructions are the same as in Palanisamy et al. (2012): they display regional sea-level changes that are similar to the global average from 1950 to 2010. The projections correspond to the regionalization of the global values presented in Table 1. The regional global median and likely sea-level projections in Guadeloupe remain close to the global average (Oppenheimer et al., 2019). However, the high-end scenario is slightly higher in Guadeloupe than at global scale. This is due to high-end scenarios involving larger

contributions from the Greenland ice sheet surface mass balance (Furst et al., 2015) and Antarctic dynamics of marine ice sheets (Spada et al., 2013;Ritz et al., 2015;DeConto and Pollard, 2016;Edwards et al., 2019). In fact, the mass losses in these two polar regions change the Earth gravitational field and rotation in a way that sea levels rise faster in tropical regions such as Guadeloupe (Slangen et al., 2012;Slangen et al., 2014;Kopp et al., 2014).


## 4.2 Regional and local vertical ground motion

*Regional vertical ground motion*

The results obtained by the two groups of the Nevada Geodetic Laboratory (NGL) and the University of La Rochelle (SONEL) are presented in Table 2. They reveal contrasting trends, with one uplifting station, two others subsiding. However, the quality of the data is also limited, with two time series displaying discontinuities potentially due to system changes, and two others that are too short. The area is also affected by different earthquakes, including subduction earthquakes (e.g., in 1843), intraplate

events (e.g., 2004), and volcanic events, with no obvious connections with existing discontinuities in GNSS time series.

Furthermore, the tide gauge of Le Gosier is not measuring a very rapid rate of relative sea-level rise, but only a trend of 0.7mm/yr, as for the Refmar data (Table 3). This estimates comes with large uncertainties due to relatively short and scarce data, but it still suggests low vertical ground motion at the tide gauge, since the geocentric sea-level rise rates, as computed from satellite altimetry, is 1.9+/-0.9mm/yr (Palanisamy et al., 2012). Furthermore, the background vertical ground motion

computed directly from the sea-level time series is -0.05 +/- 4 mm/yr (Kopp et al., 2014), suggesting again stability, but with large uncertainties.

The effects of GIA are included in the geocentric vertical ground motion estimates from GNSS stations and in the sea-level trend directly computed from the tide gauge (0.7mm/yr), but not in the sea-level estimate from altimetry and in the background vertical ground motion from Kopp et al. (2014). These GIA effects account for a subsidence of approximately 0.18+/-0.1mm/yr

according to the two GIA models used in the IPCC. Again, these estimate are uncertain, and the error could exceed 1mm/yr, as it is based on two GIA models only (Jevrejeva et al., 2014).

Hence, we obtain contrasting trends (Tables 2 and 3), which are consistent with those estimated from previous work (Sakic et al., 2020). They may reflect different vertical motion in the area of each GNSS station, some very local vertical ground motion not monitored by InSAR, or unreliable trends due to system changes and short time series.

*Local vertical ground motion*

The multi-temporal processing of the data, carried out using the GAMMA tool, allowed us to extract two products that highlight the ground displacements on the Island of Guadeloupe and its evolution during the period observation period (2003-2010). In short, we measure changes of distances between the ground surface and the satellite at different passes over Guadeloupe. These movements are visualized in color-coded form where red represents an increasing distance from the radar target of the satellite

and blue represents a decreasing distance of the radar target towards the satellite (convention chosen by the authors). Given the configuration of the satellite and the data acquisition geometry, we conclude that the ground movement highlighted in the linear velocity map (Figure 4) could be due to subsidence-type vertical motion (red ) or swelling (blue) or projection of horizontal movement on the line of sight. Figure 4 shows a ground movement around the Bouillante bay of the order of 3-6 mm/yr over the observation period, presumably due to very slow gravitational landslides or ground subsidence. If these

movements are revealed linear over time, they will represent a major contribution to relative sea-level changes along the western coast of Basse-Terre for the decades to come. This could be further investigated in future studies, also considering the





fact that the observed signal could result from complex 3-dimensional processes, which could be characterized using both ascending and descending modes when more SAR or GNSS data will be available.

However, the area of the Petit-Cul-de-sac marin, which is the focus of this study, appears relatively stable: we find vertical

ground motion of 0.4mm+/-1mm/yr in Jarry (Baie-Mahault) with respect to Les Abymes (close to the airport, Table 2). The error is computed as one standard deviation of the vertical ground motion measured in the 35 pixels in the Jarry area. This is at the limit of the capability of the techniques, so that these vertical ground motion can be considered insignificant. Since there are no measurements from the ENVISAT platform after December 2010, this velocity estimation via linear regression should be considered carefully. In fact, non-linear ground motion due to different phenomena (water table variations, local

subsidence/uplifts) and uncompensated tropospheric delays could bias the velocity estimation on a short time of observation. Furthermore, this first estimate is affected by the filtering performed within the InSAR processing procedure. To complete this error assessment, we also calculate an upper bound for the error by dividing the typical residual error from the atmospheric fluctuations (typically 1 cm on a single interferogram at a city scale) (Williams et al., 1998), by the mean duration between two acquisitions and the square root of the number of independent interferograms (Le Mouelic et al., 2005). We find 5mm/yr

for an upper bound of errors in the InSAR based vertical ground motion maps. While this is large, this is not sufficient to explain the discrepancies between different GNSS measurements.

Hence, the uncertainties associated to the measurements of ground motion velocities range from 1mm/yr (based on the variability of observations) to 5mm/yr (maximum possible value). Given these uncertainties, we could not detect any local vertical ground motion in the area of the Petit-Cul-de-sac marin, suggesting that local increases of chronic flooding events are

unrelated to rapid (centimetric) but local vertical ground motion. Although the 7 years of the processed SAR archive is the longest available in the region, combination with other missions (such as Sentinel 1) could be considered in further works addressing other related scientific questions such as the potential motion on the eastern coast of the island.

*Scenarios for vertical ground motion*

To summarize, the vertical ground motion maps derived from InSAR suggest stability with respect to a terrestrial reference

frame in the area of Jarry (Baie-Mahault) and Pointe-à-Pitre, within the error of the InSAR technique. Hence, we reject the hypothesis that each individual GNSS or tide gauge instrument considered above is measuring different vertical ground motion that are representatives of some local processes such as sediment compaction linked to variations of the groundwater contents. Therefore, the contrasting trends given by the different instruments could be due to very local processes affecting single antennas, or to discontinuities due to system changes. In fact, the trends range from -6mm/yr to 3.5mm/yr (Table 2 and 3),

which corresponds to very rapid subsidence and uplift rates. Furthermore, all measurements are scarce and include discontinuities, to the point that they are considered not robust as per the SONEL assessment. Hence, we argue that these trends are suspicious and more research is needed to understand and project vertical ground motion trends in Guadeloupe.

Based on these lines of evidence, we define scenarios for vertical ground motion in Jarry (Baie-Mahault). Similarly to what has been conceptualized to address deep uncertainties affecting climate-induced sea-level rise projections (Stammer et al.,

2019), we define two scenarios for future vertical ground motion in Guadeloupe:



- Vertical ground motion essentially due to GIA effects, that is, of 0.18+/-0.1mm/yr of subsidence, based on an assumption of no significant vertical ground motion and the computation of errors in Jarry area (Table 2; Figure 3.A)
- A "high-end" subsidence scenario of 2.3mm/yr, corresponding to a possible regional subsidence of 2mm/yr superimposed on a GIA-induced subsidence of 0.3mm/yr, which is the upper bound of GIA effects according to the IPCC projections (Table 2; Figure 3.B).

Furthermore, although we consider linear rate of vertical ground motion, we recognize that it may occur non-linearly, for example in case of tectonic or volcanic event.

## 4.3 Total water levels causing chronic flooding

Figure 5 shows the tidal signal as analyzed following the method described in subsection 3.3. This figure displays the cyclonic events as blue lines, which we further highlight in red where these events affect our dataset. Cyclone-induced storm surges can reach several tens of centimeters at Pointe-à-Pitre (e.g. ~40 cm for the David cyclone, 1979) while the modeled 100 year return period storm surge, including cyclones, is about 1.15m (Krien et al., 2015). For example, the cyclone that induced the strongest flood over the period of interest was Hugo (1989, first blue line on Figure 5).

The daily maxima of total water levels that we consider are not only caused by tidal variations, but also by non-cyclonic surges and other processes causing seasonal to interannual sea-level variations. Overall, the amplitude and recurrence of these phenomena falls within the range of typical high-water level events that can be classified as chronic flooding events (Figure 5). Hence, once removed from the cyclone events, we obtain a distribution of highest daily water levels, which are representative of moderate conditions. For example, the largest recorded water levels over the 1983-2016 does not correspond to a cyclone, but to a seasonal high monthly mean sea-level record. Hence, the distribution of daily high water levels is suitable for the study of chronic flooding, driven by tides, seasonal variations of mean sea levels and non-cyclonic surges.

## 4.4 Exposure of coastal sites

Table 4 shows the altitude of vulnerable coastal sites identified by Boudon and Chiozzoto (2012) based on LiDAR LITTO3D© data. This analysis shows that the median altitude of high vulnerability sites is 0.8m in the local reference frame (IGN88). Medium and high vulnerability sites have median altitudes of 1m and 2m in the local reference frame respectively. Some individual altitudes are doubtful, suggesting errors in geo-referencing of the coastal site or in the LiDAR dataset itself, as discussed in section 3.2. Due to these suspicious values, we consider here only the median altitude of each category of vulnerable coastal site.





## 4.5 Synthesis: past and future chronic flooding hazards for vulnerable coastal sites

The number of submersions per day from 1950 to 2100 is presented in Figure 6 for the three idealized vulnerable coastal sites

and for the two subsidence scenarios. The results show that depending on the actual altitude of each site, a rapid increase in the number of flood days per year is expected to take place sooner or later during the second half of the 21st century. This is consistent with previous work undertaken in other coastal sites (e.g., Sweet and Park, 2014), and it can be explained by the projected acceleration of sea-level rise after 2050 under RCP8.5.

In the remaining of this paper, we discuss to which extent we can attribute observed chronic flooding to sea-level rise (section

5.1), what are the times of emergence of chronic flooding events depending on the scenario considered (section 5.2) and the implications for coastal management (section 5.3) and the limitations of our approach (section 5.4).

## 5 Discussion

### 5.1 Attribution of observed chronic flooding

Whatever the scenario considered, Figure 6 shows that chronic flooding are not expected to emerge earlier than 2030 for an

altitude of 0.8 m (reference frame: IGN88) corresponding to an average high vulnerability coastal sites. However, the LiDAR data show that some coastal sites have altitude below this value (Table 4). For example, if we consider an altitude of 0.5 m, the model predicts one day of chronic flooding after the 1990s (Figure 7). However, there are also coastal sites where chronic flooding has been observed, although the altitude according to the LiDAR dataset is well above 1 m (Bourdon and Chiozzotto, 2012). This suggest that observed chronic flooding (section 2) are not yet be completely driven by sea-level rise: on the

contrary, the interactions between sea levels, the groundwater table and rainfall events, which we do not model here, probably play a significant role today. Hence, our results suggest that sea-level rise may already have caused chronic flooding at coastal sites with the lowest altitudes. However, we do not formally attribute observed chronic flooding to sea-level rise alone because the lowest sites are not always those where chronic flooding has been reported. In fact, some of the chronic flooding events observed today seem to involve ground water rise (Bourdon and Chiozzoto, 2012) or stormwater runoffs as well.

### 5.2 Emergence of chronic flooding

For coastal sites above 0.8m (IGN88), the onset of chronic flooding is not projected to take place before the 2030s (Figure 6). If we rely on the SROCC sea-level scenarios, the high vulnerability sites are projected to start experiencing chronic flooding in 2050 (2040 if we assume a regional subsidence of 2mm/yr) (Figure 6). However, the number of flood days per year increases rapidly after the emergence of chronic flooding: for example, high-vulnerability coastal sites are likely to be flooded every

two days between 2060 and 2100 (between 2050 and 2070 if we assume subsidence) (Figure 6). This rapid increase of the number of flooding days will leave little time for adaptation: a few years after the onset of frequent chronic flooding at one particular coastal site, the situation becomes critical, with one flooding every two days after two decades. The reason for that is threefold: first, the low altitude and the absence of defenses in the Petit-Cul-de-sac marin; second, the variability of daily



maxima of total water levels is roughly 0.4m only (Figure 5); and third, most of the coastal sites will start experience chronic

flooding after 2050, that is, once sea-level rise has started accelerating significantly as per RCP8.5 (Figure 3).

Despite being unlikely, high-end scenario beyond the likely range can be useful information for risk averse users (Stammer et al., 2019), such as airport and harbors authorities and the electricity or hydrocarbon providers in Jarry (Baie-Mahault) (Figure 1). For our high-end scenario, chronic flood events driven by sea-level rise occur one decade earlier than for the upper bound of the likely range. Furthermore, once the process is initiated, chronic flood events happen every day one decade after their

emergence. Therefore, as expected, the high-end scenario leaves even less time for adaptation than the baseline SROCC-based sea-level rise projections.

### 5.3 Potential measures to manage future chronic flooding events

A first measure to prevent the future impacts of chronic would be to limit sea-level rise by reducing greenhouse gas emissions. By doing so, chronic flooding would not be avoided in the locations identified as vulnerable in the Petit Cul-de-sac marin,

because sea level will continue to rise at at least rates of 3mm/yers for decades (Oppenheimer et al., 2019). However, the impacts would emerge later and at slower rates, thus allowing more time for adaptation. Furthermore, the structural changes in the economy that are required to achieve climate goals (Rockström et al., 2017) would offer an opportunity to reconsider the location and the nature of critical infrastructures in Guadeloupe and elsewhere.

Without climate change mitigation, adaptation of coastal infrastructures will be required in the Petit Cul-de-sac marin, possibly

as early as the 2030s, regardless of evolution in the intensity or the trajectory of tropical cyclones and hurricanes for futures decades (Chauvin et al., 2020). Figure 6 shows that a small difference in the elevation of coastal areas allows to avoid chronic flooding for decades. However, small walls or dike should not be efficient in this particular case due to the interactions between rainfalls, groundwater flows and sea-level changes in the former mangrove areas, which include fine sediments and porous soils. Hence, raising the ground levels could be an efficient adaptation measure. However, this measure should be taken for a

large number of already urbanized locations, which would be a challenge.

Besides sea-level rise, other cascading impacts such as the combined effects of sea-level rise and waves changes could also affect port operability (Camus et al., 2019). However, our understanding of the coastal area is that that the most urgent climate-related challenge for Jarry (Baie-Mahault) is adaptation to chronic flooding induced by sea-level rise. Other challenges include prevention and preparedness to cyclones, heavy tropical rainfalls, tsunamis, and sustaining ecosystem services (Krien et al.,

2015;Jevrejeva et al., 2020;Chauvin et al., 2020). In particular, one important issue will be the management of storm water drainage in a context where the soil is largely impermeable due to the sprawl of commercial and industrial areas.

Overall, the situation in Guadeloupe is representative of many other tropical islands where critical infrastructures are located in low-lying-areas (e.g., Atolls islands, but also high-islands such as La Réunion, the Society islands in Polynesia, the small Antilles islands) and where extreme total water levels events have relatively small amplitudes (Oppenheimer et al., 2019). In

such areas, the increasing number of chronic flooding will become a game changer for coastal management. To better anticipate such chronic flooding and adaptation needs, it would be interesting to build upon the experience of Mayotte, where such events have emerged in 2019 after a subsidence of more than 20cm caused by the eruption of a submarine volcano off the island



(Lemoine et al., 2018;Cesca et al., 2020). In fact, Mayotte could become a natural laboratory for future chronic flooding in other tropical islands, both in terms of impacts and adaptation measures that will be taken.

**5.4 Limitations of the approach and residual uncertainties**

Our results are associated with a number of residual uncertainties: first, more research would be needed to characterize vertical ground motion in the island and potentially define more precise subsidence scenarios (section 4.2). Together with the actual rates of future sea-level changes, this local feature is probably the largest source of uncertainty relevant to our estimate of times of emergence of future chronic flooding.

Other residual uncertainties relates to the assumption that the highest water levels are simply translated upwards as sea level rises. In fact, we assume here that the potential effect of the sea-level rise on the tidal characteristics is negligible in front of the sea-level rise contribution itself. This assumption is supported by global studies showing that the Caribbean islands could be affected by an increase of M2 (and also S2 and O1) and mean high water levels (accounting only for M2, S2, K1 and O1), but in relatively small proportions (Pickering et al., 2017;Schindelegger et al., 2018). For example, the amplitude of M2, S2,

K1 and O1 would increase by about 2.5 cm, 1, 0, 1, respectively, assuming no shoreline recession at the coast, for a uniform sea-level rise of 2m, after Pickering et al. (2017). This implies that taking into account this effect would only slightly modify the projected times of emergence (Figure 6 and 7), and that the chronic flooding in the Petit-Cul-de-sac marin would increase only slightly faster than the one predicted here.

In some areas, tide gauge data can contain a contribution of the wave setup affecting all the surrounding area (regional wave

setup) or some specific ports (Thompson and Hamon, 1980;Bertin et al., 2015;Pedreros et al., 2018;Melet et al., 2018). In the Petit-Cul-de-sac marin, the wave set up contribution is excepted to be negligible both at tide gauge locations and at the coastal locations vulnerable to chronic flooding, except in during cyclones (not considered in our study). Indeed, the non-cyclonic waves come most of the time from the N to W direction (trade winds) and thus are diffracted by the Grande-Terre island and arrive at the entrance of the Petit-Cul-de-sac marin with very low energy.

Hence, to conclude, beside the research priority of better characterizing future contributions to sea-level rise, the most important residual uncertainty relevant to chronic flooding in Guadeloupe appears to be the vertical ground motions.

**6 Conclusion**

In this paper, we have characterized the times of emergence of chronic flooding due to sea-level rise in the Petit-Cul-de-sac marin, Guadeloupe, a tropical island where critical infrastructure is located in low-lying areas. While rainfall and groundwater

processes seem to play a key role chronic flooding events observed so far, the number of flood days is projected to increase drastically under RCP8.5 just a few years or, at the latest, two decades after the first flood event has occurred. Depending on the actual altitude of the site considered and the regional subsidence of the island, this may occur by 2040 or later during the 21[st] century. Subsidence of the island remains a critical unknown, which we address here using vertical ground motion





scenarios, similarly to what is done for climate-induced sea-level rise. In the case of Guadeloupe, the uncertainty of subsidence
makes a large difference in times of emergence of chronic flooding. This topic would deserve more research.

The rapid increase in the number of days with chronic flooding can be explained by three factors: the low altitude and the absence of defenses in the Petit-Cul-de-sac marin, the small amplitude of tides, and the rapid increase of sea-level rise during the second half of the 21$^{st}$ century. While raising the ground levels by a few tens of centimeter allows to buy time for adaptation, this issue of chronic flooding remains a challenge due to the number of vulnerable locations and the interaction with rainfall
and groundwater processes, that make diking a barely viable option. Limiting greenhouse gas emissions would not only allow to buy time, but it would also require large transformation of the infrastructure in place. This offers an opportunity to design new infrastructure in a way that it is less vulnerable or less exposed to sea-level rise.

Far from being isolated, the case of Guadeloupe is representative of many small islands, with critical infrastructure located in low-lying area. Locally, adaptation needs seem manageable. However, the number of places that are exposed to chronic
flooding (i.e. most small island with critical low-lying areas) raises the need for a more global strategy for public authorities and infrastructure managers such as harbors, airports and other businesses in small islands.

**Acknowledgement**

This research was initiated in C3AF project with the financial support of European Regional Development Fund, Guadeloupe Regional Council and BRGM, with additional methodological inputs provided from the ANR/Storisk and ERA4CS
INSeaPTION (Grant 690462) and a FFABR 2017 (Finanziamento delle Attivita' Base di Ricerca) grant of MUR (Ministero dell' Istruzione, dell'Universita e della Ricerca) and by a research grant of Dipartimento di Scienze Pure e Applicate (DiSPeA) of the Urbino University "Carlo Bo". We thank Valérie Ballu (University of La Rochelle) for useful exchanges on vertical ground motion in the Antilles, as well as Yann Krien (University of French Antilles), Aurélie Maspataud, Erwan Bourdon and Jean-Marc Mompelat (BRGM) for discussions on chronic flooding in overseas territories. We thank the European space agency
for providing ASAR data and SONEL and NGL for providing their solutions.



**Tables**

Table 1: Assumptions for global mean contributions to sea-level changes by 2100 relative to 1986–2005 for the AR6/SROCC median and likely range (in brackets), and for our the high-end, and their implications in Guadeloupe. (See Thiéblemont et al., 2019). Note: the likely range of the sum is not equal to the sum of the likely range due to dependencies between components (Church et al., 2013a;Le Bars, 2018). We extract Guadeloupe data at the following coordinates: 61°S, 16°N. The computation of the total uses the same dependencies schemes as the regional approach of Church et al. (2013), which, together with the
choice to remove MIROC-ESM and MIROC-ESM-CHEM models, accounts for differences with the data published in SROCC (Oppenheimer et al., 2019).

| Component | RCP8.5 IPCC AR6/SROCC (excluding MIROC-ESM and MIROC-ESM-CHEM for sterodynamic effects) | RCP8.5 IPCC AR6/SROCC Guadeloupe | High-End | High-End Guadeloupe |
|---|---|---|---|---|
| Sterodynamic effects | 0.30 [0.18 to 0.42] m | 0.30 [0.23 to 0.37] m | - | 0.37 m (based on worst model) |
| Glaciers | 0.18 [0.10 to 0.26] m | 0.17 [0.10 to 0.25] m | 0.29 m | 0.27 m |
| Greenland ice sheet (surface mass balance) | 0.1 [0.04 to 0.22] m | 0.09 [0.04 to 0.20] m | 0.23 m | 0.21 m |
| Greenland ice sheet (dynamic effects) | 0.05 [0.02 to 0.09] m | 0.05 [0.02 to 0.07] m | 0.11 m | 0.09 m |
| Antarctic ice sheet (surface mass balance) | −0.05 [−0.09 to −0.02] m | −0.05 [−0.09 to −0.02] m | 0.0 m | 0.0 m |
| Antarctic ice-sheets dynamic effects | 0.16 [0.02 to 0.37] m | 0.20 [0.02 to 0.46] | 0.80 m | 0.99 m |
| Groundwater | 0.05 [−0.01 to 0.11] m | 0.048 [-0.016 to 0.11] m | 0.11 m | 0.11 m |
| GIA | - | 0.018 [0.005 to 0.031] m | - | 0.031 m |
| Total | 0.80 [0.52 to 1.16] m | 0.82 [0.55 to 1.17] m | - | 2.07 m |





Table 2: Trends obtained from the two GNSS stations located close to Jarry.

| GNSS station | Solution of the Nevada Geodetic Laboratory (NGL) | Solution of the University of La Rochelle (Sonel) |
|---|---|---|
| ABMF (Les Abymes) | -5.7+/-1.9 mm/yr | -4.2+/-0.2 mm/yr Not robust |
| FFE0 (Fort Port D'Epée) | 3.5+/-1.6 mm/yr | Not computed: not robust |
| Pointe-à-Pitre PPTG | -3.7+/-3.3 mm/yr | Not computed: too short |
| Le Gosier | -0.3+/-1.8 mm/yr | Not computed: too short |





Table 3: Pointwise vertical ground motions estimates from different sources.

| Source | Reference frame | Estimate | Residual uncertainties |
|---|---|---|---|
| Global isostatic adjustment models (GIA) | Geocentric | 0.18+/-0.1 mm/yr, based on two GIA models, as in Church et al. (2013): | Regional constraints on GIA models (Jevrejeva et al., 2014) |
| GNSS stations | Geocentric | From -5.7 to 3.5+/-1.9 mm/yr (Santamaria-Gomez et al., 2017;Blewitt et al., 2018) | Interventions on the devices, earthquakes |
| Satellite altimetry mean sea-level rise | Geocentric | 1.9+/-0.9 mm/yr (Palanisamy et al., 2012) | Regional residual uncertainties as large as 1-2 mm/yr (Ablain et al., 2015) |
| Trend at the Pointe-à-Pitre tide gauge | Local terrestrial | 0.7 mm/yr (see subsections 3.3 and 4.3) | Short time series (19 years) with gaps |
| Background subsidence at the Pointe-à-Pitre tide gauge | Local terrestrial | -0.05 +/- 4 mm/yr (Kopp et al., 2014) | Short time series (19 years) with gaps |






Table 4: altitude of coastal locations according to the LiDAR altitude dataset (reference: regional geodetic reference, IGN88).

| Exposure to chronic flooding (after Boudon and Chiozzoto, 2012) | Mean | Median | Minimum value | Maximum value | Standard deviation |
|---|---|---|---|---|---|
| Low | 2.3 m | 2.0 m | 0.6 m | 6.8 m | 2.0 m |
| Medium | 1.2 m | 1.0 m | -0.2 m | 2.4 m | 0.8 m |
| High | 0.8 m | 0.8 m | -0.3 m | 2.1 m | 0.7 m |




**Figures**

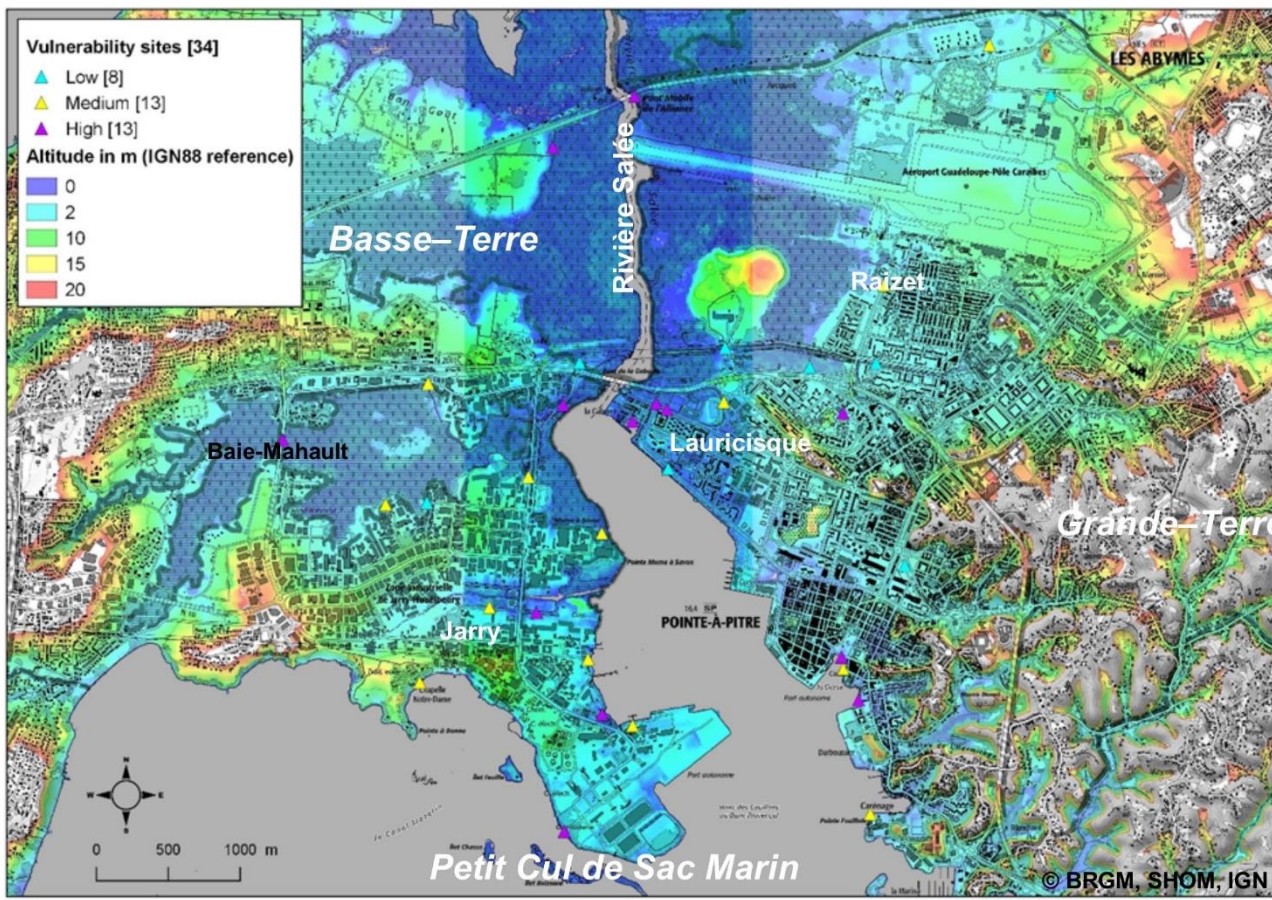

Figure 1: The Petit-Cul-de-sac marin area in Guadeloupe, showing urbanised areas and altitudes (Map created by BRGM; Data: IGN, SHOM; © BRGM, IGN, SHOM).





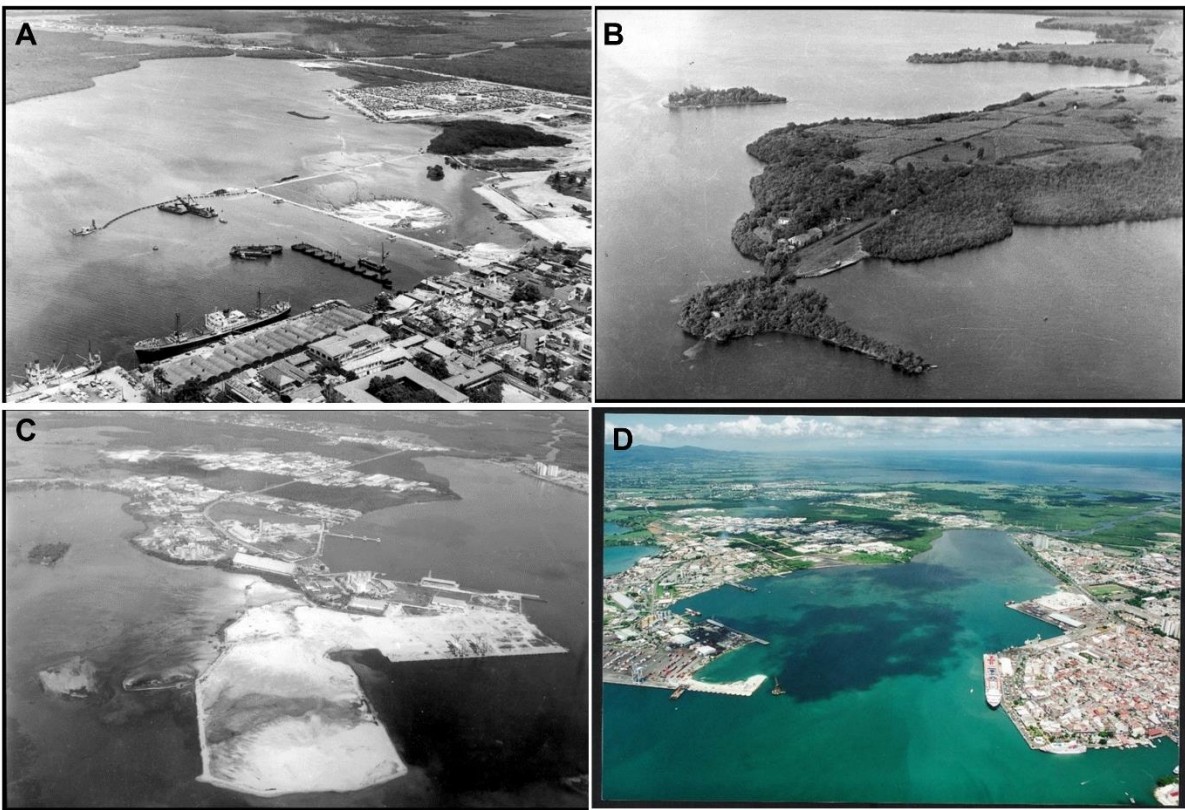

Figure 2: urbanization of the Petit-Cul-de-sac marin area since 1960 (source : CPMG) : A : construction of Bergevin embankments in Pointe-à-Pitre in 1960, involving 20 hectares of land reclamation; B : Jarry before harbor development in 1964; C: Land reclamation in Jarry in 1970; D: Jarry and Pointe à Pitre in the 1990s.


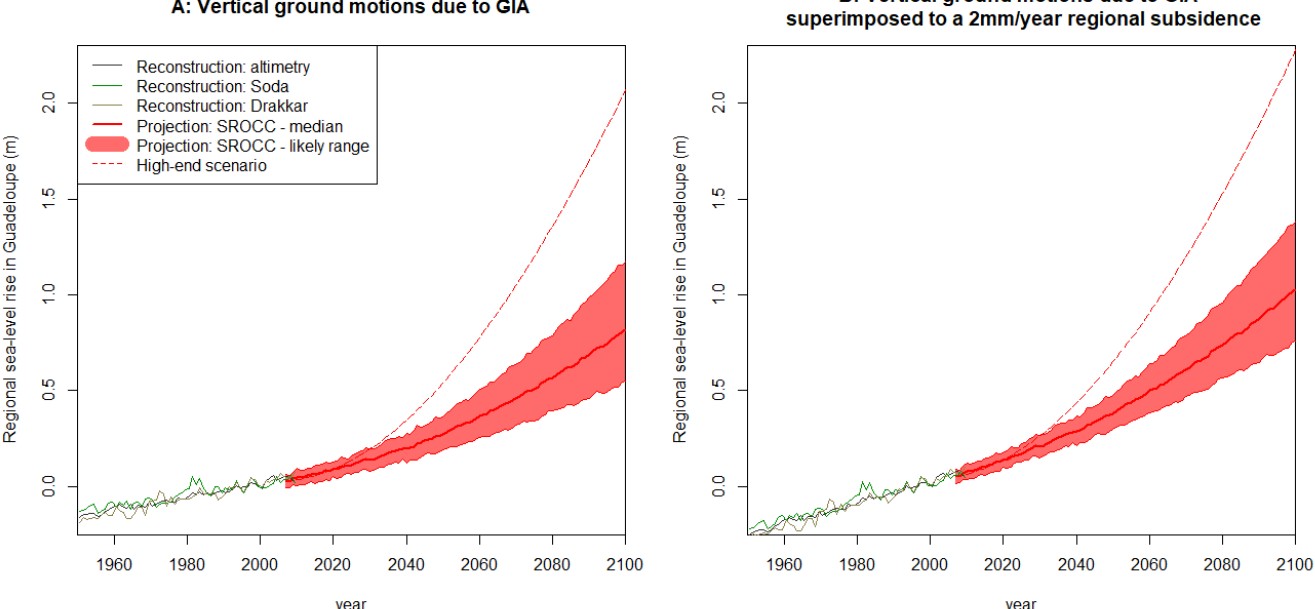

Figure 3: sea-level reconstructions and projections for Guadeloupe, with respect to 1986-2005: A: assuming vertical ground motion are due to GIA only; B: assuming an additional regional subsidence of 2 mm/yr. The three reconstructions refer to those presented in Palanisamy et al. (2012). The projections are regional implications of the Special Report on the Ocean and Cryosphere in a changing Climate (Oppenheimer et al., 2019). The high-end is the regional implication of the high-end projections presented in Thiéblemont et al. (2019).
Figure 4: linear velocity map from the multi baseline InSAR method. These results are based on the analysis of 34 radar scenes
acquired between 2003 and 2010 by the ASAR sensor on ENVISAT of the European Space Agency (Map created by BRGM;
Data: IGN, ESA; © BRGM, IGN, ESA).





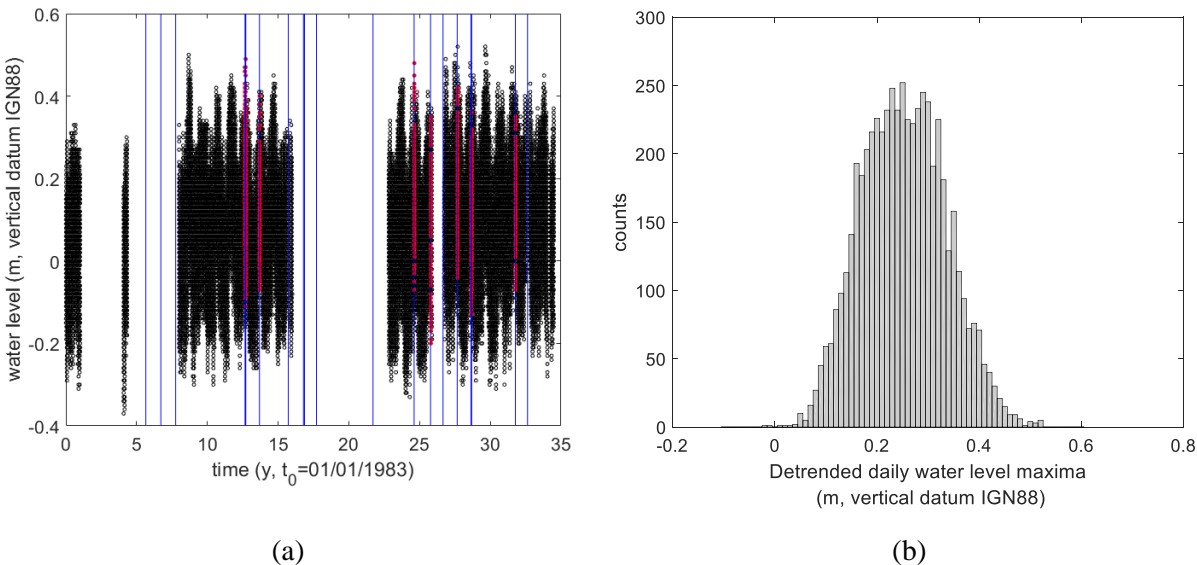

(a)                  (b)

Figure 5: tide gauge data. (a) time series of raw data (black: raw cleaned data, red: data removed from the analysis, and corresponding to the cyclones, identified with blue lines). (b) distribution of daily maxima, after the water level data post-

processing described in section 3.3 (cyclone removed, year and day filtering, detrended sea-level rise).



Figure 6: reconstruction and projections of chronic flooding events, for two subsidence scenarios and three idealized types of coastal sites in Jarry and Pointe-à-Pitre. The color code is the same as in Figure 3.


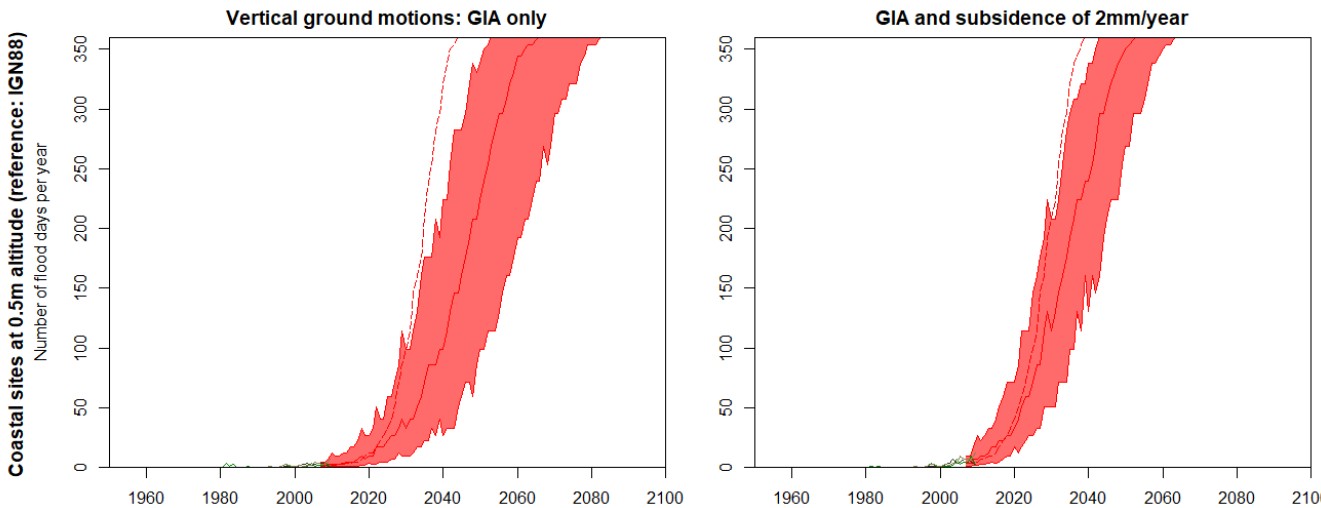

Figure 7: reconstruction and projections of chronic flooding events, for two subsidence scenarios and a coastal site located at an altitude of 0.5 m (local reference frame, IGN88). The color code is the same as in Figure 3.





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
