# Peer review of "Timescales of emergence of chronic flooding in the major economic centre of Guadeloupe."

_Natural Hazards and Earth System Sciences, 2020_

## Referee Comment (RC1) · Patrick Nunn (Referee) · 17 Sep 2020

Review of

Timescales of emergence of chronic nuisance flooding … in Guadeloupe

For 'Natural Hazards and Earth System Sciences'

This is an excellent paper, well worth publishing.  Its main strength is in showing how precise data about specific-site futures can be obtained and used for planning purposes.  This approach deserves to be much emulated.

I have many small comments that should be addressed in revision.

**Title**: I don't know why 'nuisance' is in the title – it is not mentioned in the Introduction – in fact not until section 3.5, then it is not defined.  Also is "chronic nuisance" a contradiction?  Should it be "chronic/nuisance"?

Line 19 – lowest not smallest

Line 20 – is not are – maybe the entire manuscript would benefit from being read for clarity.

Line 21 – this However can be substituted for Yet – However is a clumsy word – the text would benefit from having the word However taken out wherever it is used.

Line 69 – island names are repeated

Line 88 – seem not seems

Line 127 – delete height

Line 129 – this sentence is fine but add the point that such 'conversion' is highly unlikely to happen

Line 175 – identify not upraise

Line 183 – perhaps not presumably

Line 221 – datasets

Line 228 – change 'meter' to 'horizontal meters'

Line 233 – define nuisance flooding

Line 254 – will everyone know what subduction earthquakes are?  Perhaps add 'low-angle thrust' in brackets?

Line 326 – I don't understand how the 100-year return figure can be helpful in a subsiding context – surely the point is that the 100-year surge will now become a 20-year one?

Line 349- remainder

Line 364 – some more information about the groundwater rise and stormwater runoff would be helpful

Line 383 – chronic flooding

Line 385 – years

Line 395 – explain why this should be a challenge (don't just imply it will)

Line 405 – flooding events – and 'challenge' not 'game changer'

Line 429 – what are these non-cyclonic waves?  Tsunamis?  Elaborate

Line 436 – change 'just' to 'anything between'

Line 436 – delete regional

Line 442 – before rapid add 'expected'

Line 443 – change 'centimeter allows to buy' to 'centimeters buys'

Line 448 – what sort of new infrastructure?  More details would be of interest to readers.  What about floating port facilities?

Line 449 – change area to areas and add citation to (Kumar and Taylor 2015)

Kumar, L., and S. Taylor. 2015. "Exposure of coastal built assets in the South Pacific to climate risks." *Nature Climate Change* 5 (11):992-+.

---

## Referee Comment (RC2) · Anonymous Referee #2 · 30 Sep 2020

The paper addresses future increase in flood risk in locations where flooding is currently rare and infrastructure is therefore built close to sea-level, using Guadeloupe in the Caribbean as a detailed case-study. On this island as with several others, inland areas are very steep and challenging for building, so much development has focussed on very low-lying areas which were formerly mangrove. The paper focusses on nuisance flooding, ie that due to predictable high tides in calm weather, rather than hurricane-related extreme water levels.

The paper would benefit from a little rearrangement, some improvement of figures, and a thorough copy-edit for English but is otherwise good. A general suggestion - this is a

specific case study, but can other islands adopt the methodology directly? Is the code available for immediate reuse with simple substitution of the location? Where in the world would this be directly applicable?

Minor suggestions:

I think the paper may be more simply laid out if you discussed the geography, defined the 4 cut-offs (0.5m, 0.8m, 1.0m, 2.0m) , discussed which sites these encompass, then just stuck to these heights?

line 125 is effectively "since we're talking about protecting an airport, we are inherently in RCP8.5 scenario, as a lower scenario would involve changing this infrustructure anyway!" - an interesting take!

line 205 I'm not familiar with this method, I'm trusting you here.

Fig 4: GNSS results vs INSAR - it would be good to plot these together if possible - could you overlay the numbers from Table 2 on Fig 4 so we can see it in context? Recommend sticking to mm/yr or cm/yr throughout the paper, try not to mix units.

318 Label the scenarios A, B as in subsequent figures.

325 Is the flooding associated with cyclones also related to waves & high rainfall? It won't affect your results if you're taking them out but might be worth noting.

And what about the chronic flooding? If (line 365) there is already chronic flooding, how high above the tidal height those days did this occur? At which sites? How often? Does this mean you need to allow say 40cm for rainwater? Or more?

370 "every two days between 2060 and 2100" be careful here. Do you mean, every other day, or every day for half the year, or every spring tide, or something else? It might make quite a bit of difference to adaptation policy.

377 "For our high-end scenario, chronic flood events driven by sea-level rise occur one decade earlier than for the upper bound of the likely range. (dotted line on fig 6)

[Figure]

Fig 1 According to your map Fig 1, substantial areas are at 0m (should this actually be labelled 0-2m?) and would therefore be underwater more than half the time already. (Fig 5). Or is there a datum error? Local TG at chart datum? Since 0.8m is used as a key cut-off, it would be very useful to have a colour boundary on the map at this height, also at 1m.

Fig 5 - can you indicate on the figure when flooding starts at some location? Fig 5a please replot with years labelled.

Fig 7: Isn't this figure effectively just an extra (top) panel in Fig 6? Why not keep it together for simplicity? Fig 6 & 7 I suggest adding the caption from fig 3 so the lines are labelled on the same plot, there is plenty of space & it will enable easier reuse of the figure. Always better to avoid scrolling though pages if possible! Fig 6 : include the heights (0.8m, 1.0m, 2.0m) somewhere? Fig 6: Maybe change the x axis to start at year 2000?

Bourdon/Boudon & Chiozzoto 2012 is a key paper, but is not consistently spelt. The reference is not adequate to find it. It's some kind of local report? Is it online? Does it have a doi? Please translate the title. If it's not readily available, some of the quoted text could be more helpful, particularly the data in Table 4.... well OK, in a sense it doesn't matter, it's just that you've defined 3 sets of sites, with "Low vulnerability" as "at 2m" etc. B&C2012 is used to justify this choice? You could just stick to these numbers and maybe list some examples of each in Table 4, for the sake of interest.

Within the timescale of your paper, the high vulnerability sites may be flooded not just at high tide but over the whole tidal cycle. This seems worthy of a mention?

IPCC reference missing doi. And several others. Please check all references.

Data availability statement?

Language: The paper needs a thorough copy-edit for English. Some sentences will need significant changes, so the authors must ensure that their intended meaning is

preserved. Eg (there are many more examples!) 32 since->for 44 their->the 46 to->with etc etc

60 future mean relative sea-level? 165 artificialized - > reclaimed? 215 ...data from these days are removed... 312 suspicious -> suspect 314 Similarly to what has been conceptualized to address deep uncertainties affecting climate-induced sea-level rise projections -> In a similar manner to climate scenarios for SLR projections [ref], to address deep uncertainties we define... etc 349 which->what
* * *

---

## Author Comment (AC1) · 13 Nov 2020

First, we would like to thank Patrick Nunn for his review and apologize for responding late to his review.

**This is an excellent paper, well worth publishing. Its main strength is in showing how precise data about specific-site futures can be obtained and used for planning purposes. This approach deserves to be much emulated.**

⇨ We thank you for your appreciation of this work.

**I have many small comments that should be addressed in revision.**

**Title: I don't know why 'nuisance' is in the title – it is not mentioned in the Introduction – in fact not until section 3.5, then it is not defined. Also is "chronic nuisance" a contradiction? Should it be "chronic/nuisance"?**

⇨ We agree that nuisance is not necessary in the title. In fact, the literature is using different terms for the same phenomenon (high-tide flooding, chronic flooding and nuisance flooding). It seems to us that chronic flooding is appropriate. The term "nuisance" flooding makes assumptions on the impacts for human activities. This has been corrected in the title, lines 27 and lines 233.

**Line 19 – lowest not smallest**

⇨ Thank you. This has been corrected.

**Line 20 – is not are – maybe the entire manuscript would benefit from being read for clarity.**

⇨ Tank you. This has been corrected. The entire manuscript is being proofread by all coauthors to minimize English errors.

**Line 21 – this However can be substituted for Yet – However is a clumsy word – the text would benefit from having the word However taken out wherever it is used.**

⇨ Thank you. The word "however" is used approximately 20 times in the discussion paper and is nowhere necessary. We have rephrased these sentences.

**Line 69 – island names are repeated**

⇨ Thank you and sorry about this mistake

**Line 88 – seem not seems**

⇨ Thank you. Done.

**Line 127 – delete height**

⇨ We agree. Thank you.

**Line 129 – this sentence is fine but add the point that such 'conversion' is highly unlikely to happen**

⇨ We agree that there is currently no strong sign that this transformation is being initiated. We have mentioned this point and cite Nachmany and Mangan (2018) to support this statement. Nachmany, M.; Mangan, E. *Aligning National and International Climate Targets*. 2018. Available online: **http://www.lse.ac.uk/GranthamInstitute/publication/targets/** (accessed on 04 November 2020).. The sentence now reads: "Yet, there is no strong signal that such a transformation is being initiated (Nachmany and Mangan, 2018). Furthermore, we do not

know how future energy and transportation infrastructures will look like after such a transformation. This makes any assessment of their vulnerability highly speculative."

**Line 175 – identify not upraise**

⇨ Thank you: here we have used the term "evaluate" because the techniques provide some quantitative information (although with large uncertainty).

**Line 183 – perhaps not presumably**

⇨ Thank you, done.

**Line 221 – datasets**

⇨ Thank you, done.

**Line 228 – change 'meter' to 'horizontal meters'**

⇨ Thank you, done.

**Line 233 – define nuisance flooding**

⇨ Nuisance flooding is used in several papers addressing the same problem of high-tide flooding (e.g., Jacobs et al., 2018; Moftakari et al., 2015, 2017). However as discussed above we agree with the reviewer that the term 'chronic flooding' is more appropriate in the context of our study. We have changed the sentence accordingly.

**Line 254 – will everyone know what subduction earthquakes are? Perhaps add 'low-angle thrust' in brackets?**

⇨ We agree and included 'low angle thrust' in brackets after "subduction earthquake".

**Line 326 – I don't understand how the 100-year return figure can be helpful in a subsiding context – surely the point is that the 100-year surge will now become a 20-year one?**

⇨ We agree that this can be confusing: the Krien et al. study has modelled surge and wave setup from a large datasets of cyclones. However, it applies for present-days bathymetry and sea-levels. We have removed this reference and only mention observations of past cyclones (Hugo, 1989 and David, 1979).

**Line 349- remainder**

⇨ Thank you

**Line 364 – some more information about the groundwater rise and stormwater runoff would be helpful**

⇨ Thank you for this comment. We have investigated further the technical/engineering literature to precise this issue, with the additional help of our colleague Benjamin Seux, hydrogeologist (now cited in acknowledgements). First it appears that there is few information on the impacts of sea-level rise for groundwater salinization and the groundwater levels. A report published in 2011 (in French) provides the state of the knowledge on this issue i.a. in Guadeloupe, and we are not aware that more precise observations or modelling work has been done since the publication of this report. https://www.documentation.eauetbiodiversite.fr/notice/influence-de-la-montee-du-niveau-de-la-mer-sur-le-biseau-salin-des-aquiferes-cotiers-des-drom0. This report is not conclusive on the topic of groundwater levels. Hence, we just note, as noted by

Bourdon and Chiozzotto (2012) already, that changing groundwater levels may play a role (as expected on a former mangrove). We also note that the aquifer we are considering in our study is probably not a priority in terms of hydrogeological investigations, as others are more critical for water resources management.

⇨ We found some further evidences (DEAL, 2015) suggesting that rainfall and water runoff should play a role in the observed phenomena. These phenomena take place not only during cyclones but also during seasonal heavy rainfall events and can temporarily challenge water drainage systems. They affect primarily urbanized area such as our sites of interest, where soil sealing prevent water from infiltrating the ground. We added these precisions in the manuscript.

DEAL. 2015 – Cartographie du territoire à risque d'inondation important (TRI) – Centre Guadeloupe. Rapport de présentation,[Mapping territories at risk of important innundation – Guadeloupe Center – presentation report] 53 pages.Available : http://www.guadeloupe.developpement-durable.gouv.fr/IMG/pdf/20150400_tricentre_i.pdf (accessed 11/11/2020)

**Line 383 – chronic flooding**

⇨ Thank you - corrected

**Line 385 – years**

⇨ Thank you corrected

**Line 395 – explain why this should be a challenge (don't just imply it will)**

⇨ We now precise that raising ground levels in a number of places simultaneously could be a challenge for port maintenance operations due to limited resources.

**Line 405 – flooding events – and 'challenge' not 'game changer'**

⇨ Done, thank-you.

**Line 429 – what are these non-cyclonic waves?  Tsunamis?  Elaborate**

⇨ Here we are referring to seasonal waves and how they may affect tide gauge measurements through a wave-setup. We precise now precise that the energy of seasonal waves is too small in the area of interest to significantly affect the tide gauge. Yet, it is true that tsunami risks are a reason of concern in this area. To further support this statement, we now refer to another technical report (Pedreros et al., 2007).

*Pedreros, Rodrigo ; Terrier, Monique ; Poisson, Blanche (2007) - Tsunamis : Etude de cas au niveau de la côte antillaise française. Rapport de synthèse. BRGM/RP-55795-FR, 72 p., 8 ph.*

**Line 436 – change 'just' to 'anything between'**

⇨ Thank you – the sentence has been changed to: "the number of flood days is projected to increase drastically under RCP8.5 at the latest two decades after the first flood event has occurred"

**Line 436 – delete regional**

⇨ Done thank you.

**Line 442 – before rapid add 'expected'**

⇨ Done thank you.

**Line 443 – change 'centimeter allows to buy' to 'centimeters buys'**

⇨ Done thank you

**Line 448 – what sort of new infrastructure?  More details would be of interest to readers.  What about floating port facilities?**

⇨ In fact, we believe that the most obvious example here can be the diesel thermal electricity plan, which is located in Jarry and could be replaced be renewable energy production in other areas. We have provided this example. We also understand that the suggestion of floating ports is relevant to consider as there are already floating embankments for ships up to 35m ling in the harbor of Guadeloupe. However, we have no expertise in this area and we are not sure that it can perform well for the activities in the port of Guadeloupe. Therefore, we have not included it as an example.

**Line 449 – change area to areas and add citation to (Kumar and Taylor 2015)**

⇨ We agree that this reference is relevant here, and we added it. Thank you for raising our attention to it.

Kumar, L., and S. Taylor. 2015. "Exposure of coastal built assets in the South Pacific to climate risks." Nature Climate Change 5 (11):992-+

---

## Author Comment (AC2) · 13 Nov 2020

Similarly, we would like to thank the reviewer for her/his review, and to apologize for submitting late our response. Attached is our response to the points raised by Reviewer 1. We hope it responds adequately to her/his concerns. On behalf of the coauthors Gonéri Le Cozannet, BRGM, 13/11/2020

Please also note the supplement to this comment:
https://nhess.copernicus.org/preprints/nhess-2020-178/nhess-2020-178-AC2-supplement.pdf

[Figure]

**Supplement:**

First, similarly to Reviewer 1, we would like to apologize for responding late to Reviewer 2 comments.

**The paper addresses future increase in flood risk in locations where flooding is currently rare and infrastructure is therefore built close to sea-level, using Guadeloupe in theCaribbean as a detailed case-study. On this island as with several others, inland areas are very steep and challenging for building, so much development has focused on very low-lying areas which were formerly mangrove. The paper focusses on nuisance flooding, ie that due to predictable high tides in calm weather, rather than hurricane related extreme water levels. The paper would benefit from a little rearrangement, some improvement of figures, and a thorough copy-edit for English but is otherwise good.**

⇨ We thank you for your assessment and hope that our changes to the manuscript will respond adequately to your comments.

**A general suggestion - this is a specific case study, but can other islands adopt the methodology directly? Is the code available for immediate reuse with simple substitution of the location? Where in the world would this be directly applicable?**

⇨ We will provide the code as a supplementary material. However, the difficulty is not to develop the code computing the number of chronic flooding events, but rather the availability of tidal, altimetric and vertical ground motions data. Furthermore, knowledge on the infrastructure at risk is required as well. For these reasons, we believe that the key criterion of success for adapting this study in other contexts is local knowledge. The missing piece of information for research teams having access to this local information may be the sea-level projections. Hence, we also now precise in a "data availability statement" that the sea-level projections we are using are based on those available from the University of Hamburg (https://icdc.cen.uni-hamburg.de/en/ar5-slr.html) and can be downloaded for other locations from here: https://sealevelrise.brgm.fr/).

**Minor suggestions: I think the paper may be more simply laid out if you discussed the geography, defined the4cut-offs(0.5m, 0.8m, 1.0m, 2.0m), discussed which sites these encompass, then just stuck to these heights?**

⇨ Thank you for the suggestion. We have considered changing Figure 1 to better highlight these cut-offs, but the map becomes too heavy. Therefore, we have changed the legend of Figure 1 as recommended and added an additional Figure to better identify low lying elevations and coastal sites. This also allows responding to an other comment below on Table 4. Regarding the suggested reorganization, we think that the narrative should start from exposed assets, the currently observed phenomena and the current perception of their vulnerability (Based on Bourdon and Chiozzotto, 2012; see section 2 and Figure 1). This allows later to discuss the attribution of these effects and to suggest that the altitude is not the unique criterion to be taken into account (section 5.1).

**line 125 is effectively "since we're talking about protecting an airport, we are inherently in RCP8.5 scenario, as a lower scenario would involve changing this infructure anyway!" - an interesting take!**

⇨ Thank-you. In fact we do not only mention the airport here but all the energy and transport infrastructure. The paper we quote here (Rockström et al., 2018) describe a pathway toward 1.5°C (or 2°C). It clearly highlights that such pathways require a major transformation of energy and transport infrastructure. We also note in section 5.3 that such a transformation could be seen as an "opportunity to reconsider the location and the nature of critical infrastructures in

Guadeloupe and elsewhere", and therefore reduce exposure to coastal hazards induced by committed sea-level rise. We also added, based on Reviewer 1 recommendation a note to remind that only a few country have set up the measures to achieve their own climate objectives so far.

**line 205 I'm not familiar with this method, I'm trusting you here.**

⇨ Thank you : this method of « Small Baseline Subset » is in fact a technical detail worth to be mentioned to ensure reproducibility of our study. This processing has been implemented by our coauthors Marcello De Michele and Daniel Raucoules. It is one of the procedures commonly used in InSAR processing, particularly when there are not enough images to perform Persistent Scatterers interferometry.

**Fig 4: GNSS results vs INSAR - it would be good to plot these together if possible - could you overlay the numbers from Table 2 on Fig 4 so we can see it in context?**

⇨ Thank you for the suggestion. In this case, we feel more comfortable with presenting the InSAR results without the GNSS velocities from Table 2. Unlike InSAR results, GNSS velocities have low confidence, as shown by the large errors of the NGL solutions and the "not robust" caveat of the SONEL solution (Table 2). Furthermore, our InSAR results clearly show that differences among GNSS velocities are either due to very local processes affecting single antennas, or to discontinuities due to system changes (see discussion in section 4.2). In fact, these differences motivate us considering two subsidence scenario.

**Recommend sticking to mm/yr or cm/yr throughout the paper, try not to mix units.**

⇨ We agree and modified the text and the figure accordingly. Overall we limited us to using "meters" and "mm/yr" to avoid confusion, except for tidal variations where we use millimeters (one instance).

**318 Label the scenarios A, B as in subsequent figures.**

⇨ Thank you. We are redoing Figure 6 and 7 for a consistent labelling of scenarios across figures.

**325 Is the flooding associated with cyclones also related to waves & high rainfall? It won't affect your results if you're taking them out but might be worth noting. And what about the chronic flooding? If (line 365) there is already chronic flooding, how high above the tidal height those days did this occur? At which sites? How often? Does this mean you need to allow say 40cm for rainwater? Or more?**

⇨ We agree that section 4.3 was confusing as it was unclear what data was used as input (tidal records in Figure 5.A) and what data is used to chronic flooding (Figure 5.B). The Figure 5.B excludes cyclonic events and are the total water levels used to compute the number of chronic flood events per year in section 4.5. We hope these precisions clarify the message here. The new subsection 4.3 reads as follows: '*Figure 5.A shows the raw tidal signal, and Figure 5.B shows the distribution of total water levels maxima obtained following the method described in subsection 3.3. Figure 5.A displays the cyclonic events as blue lines, which we further highlight in red where these events affect our dataset. Cyclone-induced storm surges can reach several tens of centimeters at Pointe-à-Pitre (e.g. ~0.4m for the David cyclone, 1979). the first blue line on Figure 5 corresponds to the cyclone that induced the strongest flood over the period of interest (Hugo, 1989).*

*The daily maxima of total water levels shown in Figure 5.B are not only caused by tidal variations, but also by non-cyclonic surges and other processes causing seasonal to interannual sea-level variations. Overall, the amplitude and recurrence of these phenomena falls within the range of typical high-water level events that can be classified as chronic flooding events (Figure 5). For example, the largest water level record over 1983-2016 corresponds to a seasonal high monthly mean sea-level record. Hence, once removed from the cyclone events, we obtain a distribution of highest daily water levels, which are representative of moderate conditions. Hence, the distribution of daily high water levels is suitable for the study of chronic flooding, driven by tides, seasonal variations of mean sea levels and non-cyclonic surges.'*

**370 "every two days between 2060 and 2100" be careful here. Do you mean, every other day, or every day for half the year, or every spring tide, or something else? It might make quite a bit of difference to adaptation policy.**

⇨ Thank you. We agree this was not clear and we changed to "180 days per year".

**377 "For our high-end scenario, chronic flood events driven by sea-level rise occur one decade earlier than for the upper bound of the likely range. (dotted line on fig 6)**

⇨ Thank you. We added the reference to Figure 6 as suggested.

**Fig 1 According to your map Fig 1, substantial areas are at 0m (should this actually be labelled 0-2m?) and would therefore be underwater more than half the time already. (Fig 5). Or is there a datum error? Local TG at chart datum? Since 0.8m is used as a key cut-off, it would be very useful to have a colour boundary on the map at this height, also at 1m.**

⇨ Thank you. We confirm that the datum (vertical reference) are consistent across the manuscript: we used the IGN88 local reference for both altitudes and tidal levels. We agree that Figure 1 required the suggested changes (key cut-offs and more accurate labels). We are changing the Figure accordingly. We added another Figure to help focusing on low lying elevations (see above).

**Fig 5 - can you indicate on the figure when flooding starts at some location? Fig 5a please replot with years labelled.**

⇨ Thank you. We are changing panel 5.A to have year as abscissa labels. However, we are not sure how to highlight where flood starts in particular locations. This can be identified based on the altitude on the vertical axis, as we are providing all altitudes in the same local reference frame (IGN88).

**Fig 7: Isn't this figure effectively just an extra (top) panel in Fig 6? Why not keep it together for simplicity?**

⇨ We agree that Figure 7 delivers the same information as Figure 6 for another critical height, but we propose to keep these two figures separated as the 0.5m altitude is only discussed in the context of the attribution discussion in 5.1.

**Fig 6 & 7 I suggest adding the caption from fig 3 so the lines are labelled on the same plot, there is plenty of space & it will enable easier reuse of the figure. Always better to avoid scrolling though pages if possible! Fig 6 : include the heights (0.8m, 1.0m, 2.0m) somewhere? Fig 6: Maybe change the x axis to start at year 2000?**

⇨ Thank you. We agree and are improving figures 6 and 7 accordingly. We displayed the plots over 1960-2100 to easily compare with Figures 3, but this is not necessary indeed.

**Bourdon/Boudon & Chiozzoto 2012 is a key paper, but is not consistently spelt. The reference is not adequate to find it. It's some kind of local report? Is it online? Does it have a doi? Please translate the title.**

⇨ Thank you. We changed the reference as requested, adding a link to the report. The reference now reads: „Bourdon, E., and Chiozzotto, C.: impacts géotechniques et hydrauliques de l'élévation du niveau de la mer due au changement climatique dans le contexte urbain côtier de la zone pointoise (Guadeloupe). [geotechnical and hydraulic impacts of sea-level rise caused by climate change in the urban coastal area surrounding Pointe-à-Pitre (Guadeloupe)] 135p., 2012. Available http://infoterre.brgm.fr/rapports/RP-60857-FR.pdf (Accessed 11/11/2020)."

**If it's not readily available, some of the quoted text could be more helpful, particularly the data in Table 4.... well OK, in a sense it doesn't matter, it's just that you've defined 3 sets of sites, with "Low vulnerability" as "at 2m" etc. B&C2012 is used to justify this choice? You could just stick to these numbers and maybe list some examples of each in Table 4, for the sake of interest.**

⇨ Thank you: as Figure 1 is already including a lot of information, we added another Figure is needed to help evaluating the precise altitude of each site. The vulnerability of each site was characterized by Bourdon and Chiozzotto (2012) based on Field surveys. We evaluated the altitude of all these sites and computed the statistics in Table 4 accordingly. We use the median altitude within each category as the typical value (e.g., 0.8m for high vulnerability sites). The new map helps identifying the location of each vulnerable site. The full list of sites is available in the report in the references (link to the report was added), and it is now added as a supplementary file to illustrate how we proceeded.

**Within the timescale of your paper, the high vulnerability sites may be flooded not just at high tide but over the whole tidal cycle. This seems worthy of a mention?**

⇨ Thank you. This is an important point which we have not addressed in the discussion paper indeed. This point is important to remind the consequences of doing nothing (no adaptation). The amplitude of the tide is low enough to justify considering permanent flooding over the period of interest. We are working on this point. The resulting timings of permanent flooding will be provided in section 4.5.

**IPCC reference missing doi. And several others. Please check all references. Data availability statement?**

⇨ Thank you. We have added a data availability statement and checked references.

**Language: The paper needs a thorough copy-edit for English. Some sentences will need significant changes, so the authors must ensure that their intended meaning is preserved. Eg(therearemanymoreexamples!) 32since->for44; their->the46; to->with etc etc 60 future mean relative sea-level? 165 artificialized - > reclaimed? 215 ...data from these days are removed... 312 suspicious -> suspect 314 Similarly to what has been conceptualized to address deep uncertainties affecting climate-induced sea-level rise projections -> In a similar manner to climate scenarios for SLR projections [ref], to address deep uncertainties we define... etc 349 which->what**

⇨ Thank you for these suggestions and recommendations. We have implemented all these changes and we are working to improve the use of English in our manuscript.

---

## Author Response (AR2)

Gonéri LE COZANNET
BRGM / French Geological Survey
Risk and Prevention Department
Coastal Risks and Climate Change unit
3 avenue Claude Guillemin - 45060 Orléans, France

[Figure]

Orléans, 22nd December 2020

**Subject:** **ReSubmission of "Timescales of emergence of chronic flooding in the major economic centre of Guadeloupe."**

Dear Editor,

We thank you for considering the publication of our manuscript in Natural Hazards and Earth System Sciences. On behalf of my co-authors, I am pleased to submit a revised manuscript.

We provide below a detailed response to your concerns, which we hope to have addressed satisfactorily.

We hope that this paper will be suitable for publication in Natural Hazards, and stay at your disposal for any further information.

With best regards,

On behalf of the co-authors,

Gonéri Le Cozannet

*G. Le Cozannet*

**Editor's comments**

**Thank you for the revised submission of your very interesting manuscript "Timescales of emergence of chronic nuisance flooding in the major economic centre of Guadeloupe".In the revised version, the manuscript has significantly been improved. Your research on the specific case is excellent.**

- *We thank the Editor for his appreciation of our work*

**However, before the acceptance of the manuscript, the following issues need to be resolved:**

**1. It is not explicitly described, 'why should someone outside of your study area be interested in the results'. If you were to explain the results of your case study to someone in another area, what would they gain from your case study? Do they learn from your methodology and what you encountered when applying it? What is novel and what might they learn? This (how the innovative idea of your study can be transferred to others, along with challenges) needs to be explicitly described both at the beginning so we understand, but also in discussion. This is crucial for accepting the manuscript. The existing explanation is not sufficiently satisfactory.**

- We understand the comment from the Editor and agree that this was not explicit in the previous version of the manuscript. In response to this comment, we have implemented the following changes:

- At the beginning of the abstract: we have included a sentence to remind the context of sea-level rise and climate change: "Sea-level rise due to anthropogenic climate change is not only projected to exacerbate extreme events such as cyclones and storms, but also to cause more frequent chronic flooding occurring at high tides under calm weather conditions."

- At the end of the abstract, we now write: "Similar [chronic flooding] processes are expected to take place in many low-elevation coastal zones worldwide, including in other tropical islands. The method used in this study can be transported in other locations, provided tide gauge records and local knowledge on vertical ground motions are available. We argue that identifying times of emergence of chronic flooding events is urgently needed in most low-lying coastal areas, because adaptation requires decades to be implemented, whereas chronic flooding hazards can worsen drastically within years after the first event has been observed."

- We have extended a bit the motivation of the study in the introduction; we feel this is necessary to motivate our statement that similar studies should be conducted in other regions. This reads: "Global assessments available today tend to focus on extreme events exacerbated by sea-level rise, without evaluating the significance of chronic flooding (Oppenheimer et al., 2019). Studies raising awareness on chronic flooding are mostly local, and often focus on temperate areas such as the USA and New-Zealand (Sweet and Park, 2014; Moftakhari et al., 2017;Dahl et al., 2017; Stephens et al., 2020). However, chronic flooding is also a significant matter of concern in tropical islands, because their low lying areas are critical for human activities such as trade, transport and housing (Kumar and Taylor, 2015)." We added a reference to Stephens, S. A., R. G. Bell and Lawrence, J.: Developing signals to trigger adaptation to sea-level rise. Environmental Research Letters, 13(10), 104004, doi:10.1088/1748-9326/aadf96, 2018.

- *We have extended the conclusion to provide practical recommendations: this reads:"* We argue that studies assessing future chronic flooding are urgently needed in most low-lying coastal areas across the Globe. In fact, sea-level is projected to rise along most inhabited coastlines, and adaptation takes decades to be implemented (Haasnoot et al.,2020). The case of Guadeloupe shows that adaptation should ideally be planned before the first chronic flooding events are attributed to sea-level rise. Based on our study above, we recommend that the following points are considered when performing future assessments of chronic flooding hazards:
  - Attribution of chronic flooding events is needed to assess the urgency of adaptation. Yet, the simple observation that chronic flooding events are becoming more frequent is not sufficient to formally attribute the observed phenomenon to sea-level rise. In Guadeloupe, we do not attribute formally observed events to sea-level rise due to lack of consistency between the location of hotspots and their altitude. For future study attempting formal attribution of chronic flood events, we recommend collecting and analyzing additional information on rainfall, runoff and groundwater flows.
  - The times of emergence of chronic flooding can be assessed provided tide gauge records and some information on vertical ground motions are available. Our study provides additional data and methods allowing to assess these times of emergence (see data availability statement). Where resources are limited, a simple preliminary assessment can be performed, by superimposing spring tide levels and sea-level projections available below.
  - Sea-level rise due to climate change is a major driver of future chronic flooding risks, but vertical ground motions due to natural processes or anthropogenic activities need to be considered as well for precise local assessments. In Guadeloupe, the tectonics remain a major source of uncertainties. For future study, we recommend assessing their role, using e.g. the geodetic methods used here (GNSS, InSAR), potentially supplemented with knowledge on the geology. Alternatively, in case of deep uncertainties, users may also consider following a scenario approach as presented above.
- Additional reference: Haasnoot, M. et al.: Adaptation to uncertain sea-level rise; how uncertainty in Antarctic mass-loss impacts the coastal adaptation strategy of the Netherlands. *Environmental Research Letters*, **15**(3), 034007, doi:10.1088/1748-9326/ab666c, 2020.

**2. In paragraph 420 (Discussion section), you introduced 'it would be interesting to build upon the experience of Mayotte'. ... 'Mayotte could become a natural laboratory'. How Mayotte case became suddenly important, without providing any context.**

- We agree that more context is necessary. We have extended this discussion to refer to geophysical and social observations from other case studies in the Pacific (Ballu et al., 2011; Jamero et al., 2017). We have rephrased the paragraph as follows: "To better manage adaptation to such chronic flooding events, we could learn from areas that experienced rapid relative sea-level changes due to vertical ground motions. For example, a village in the Torres Islands (Vanuatus) was relocated in 2002/2004 due to chronic flooding events caused by relative sea-level changes partly attributed to an earthquake (Ballu et al., 2011). Another earthquake that took place in 2013 in the Philippines caused the subsidence of several

islands (e.g., Batasan, Ubay) and chronic flooding events up to 135 days per year. Yet, people from these islands preferred accommodating with chronic flooding in this case (Jamero et al., 2017). Hence, the local adaptation response can be very different depending on the local context. For Guadeloupe, it would be interesting to build upon the experience of Mayotte, another French island with a similar institutional context. In Mayotte, chronic flooding events emerged in 2019 after a subsidence of about 0.2m caused by the eruption of a submarine volcano off the island (Lemoine et al., 2018;Cesca et al., 2020). The rise of sea-levels observed within two years in Mayotte is typically what is projected to take place in almost all tropical islands worldwide over the coming three decades."

- Additional references

Ballu, V., Bouin, M.N., Siméoni, P., Crawford, W.C., Calmant, S., Boré, J.M., Kanas, T. and Pelletier, B.. Comparing the role of absolute sea-level rise and vertical tectonic motions in coastal flooding, Torres Islands (Vanuatu). Proceedings of the National Academy of Sciences, 108(32), 13019-13022, 10.1073/pnas.1102842108, 2011.

Jamero, M.L., Onuki, M., Esteban, M., Billones-Sensano, X.K., Tan, N., Nellas, A., Takagi, H., Thao, N.D. and Valenzuela, V.P.. Small-island communities in the Philippines prefer local measures to relocation in response to sea-level rise. Nature Climate Change, 7(8), 581-586, 10.1038/nclimate3344, 2017.